# A New Theory for Matrix Completion

**Guangcan Liu**[*]      **Qingshan Liu**[†]      **Xiao-Tong Yuan**[‡]

B-DAT, School of Information & Control, Nanjing Univ Informat Sci & Technol
NO 219 Ningliu Road, Nanjing, Jiangsu, China, 210044
`{gcliu,qsliu,xtyuan}@nuist.edu.cn`

## Abstract

Prevalent matrix completion theories reply on an assumption that the locations of the missing data are distributed uniformly and randomly (i.e., uniform sampling). Nevertheless, the reason for observations being missing often depends on the unseen observations themselves, and thus the missing data in practice usually occurs in a nonuniform and deterministic fashion rather than randomly. To break through the limits of random sampling, this paper introduces a new hypothesis called *isomeric condition*, which is provably weaker than the assumption of uniform sampling and arguably holds even when the missing data is placed irregularly. Equipped with this new tool, we prove a series of theorems for missing data recovery and matrix completion. In particular, we prove that the exact solutions that identify the target matrix are included as critical points by the commonly used nonconvex programs. Unlike the existing theories for nonconvex matrix completion, which are built upon the same condition as convex programs, our theory shows that nonconvex programs have the potential to work with a much weaker condition. Comparing to the existing studies on nonuniform sampling, our setup is more general.

## 1 Introduction

Missing data is a common occurrence in modern applications such as computer vision and image processing, reducing significantly the representativeness of data samples and therefore distorting seriously the inferences about data. Given this pressing situation, it is crucial to study the problem of recovering the unseen data from a sampling of observations. Since the data in reality is often organized in matrix form, it is of considerable practical significance to study the well-known problem of *matrix completion* [1] which is to fill in the missing entries of a partially observed matrix.

**Problem 1.1** (Matrix Completion). *Denote the $(i,j)$th entry of a matrix as $[\cdot]_{ij}$. Let $L_0 \in \mathbb{R}^{m \times n}$ be an unknown matrix of interest. In particular, the rank of $L_0$ is unknown either. Given a sampling of the entries in $L_0$ and a 2D index set $\Omega \subseteq \{1, 2, \cdots, m\} \times \{1, 2, \cdots, n\}$ consisting of the locations of the observed entries, i.e., given*

$$\{[L_0]_{ij} | (i,j) \in \Omega\} \quad and \quad \Omega,$$

*can we restore the missing entries whose indices are not included in $\Omega$, in an exact and scalable fashion? If so, under which conditions?*

Due to its unique role in a broad range of applications, e.g., structure from motion and magnetic resonance imaging, matrix completion has received extensive attentions in the literatures, e.g., [2–13].

---

[*]The work of Guangcan Liu is supported in part by national Natural Science Foundation of China (NSFC) under Grant 61622305 and Grant 61502238, in part by Natural Science Foundation of Jiangsu Province of China (NSFJPC) under Grant BK20160040.

[†]The work of Qingshan Liu is supported by NSFC under Grant 61532009.

[‡]The work of Xiao-Tong Yuan is supported in part by NSFC under Grant 61402232 and Grant 61522308, in part by NSFJPC under Grant BK20141003.

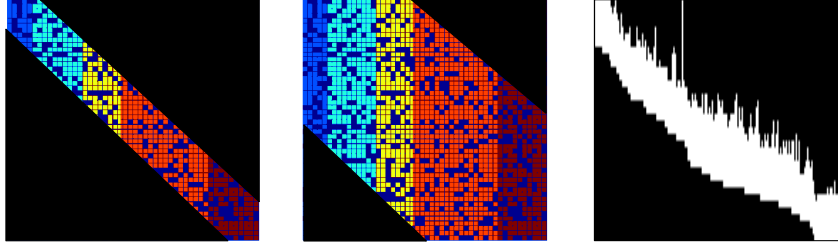

Figure 1: Left and Middle: Typical configurations for the locations of the observed entries. Right: A real example from the Oxford motion database. The black areas correspond to the missing entries.

In general, given no presumption about the nature of matrix entries, it is virtually impossible to restore $L_0$ as the missing entries can be of arbitrary values. That is, some assumptions are necessary for solving Problem 1.1. Based on the high-dimensional and massive essence of today's data-driven community, it is arguable that the target matrix $L_0$ we wish to recover is often low rank [23]. Hence, one may perform matrix completion by seeking a matrix with the lowest rank that also satisfies the constraints given by the observed entries:

$$\min_{L} \operatorname{rank}(L), \quad \text{s.t.} \quad [L]_{ij} = [L_0]_{ij}, \forall (i,j) \in \Omega. \tag{1}$$

Unfortunately, this idea is of little practical because the problem above is NP-hard and cannot be solved in polynomial time [15]. To achieve practical matrix completion, Candès and Recht [4] suggested to consider an alternative that minimizes instead the *nuclear norm* which is a convex envelope of the rank function [12]. Namely,

$$\min_{L} \|L\|_*, \quad \text{s.t.} \quad [L]_{ij} = [L_0]_{ij}, \forall (i,j) \in \Omega, \tag{2}$$

where $\| \cdot \|_*$ denotes the nuclear norm, i.e., the sum of the singular values of a matrix. Rather surprisingly, it is proved in [4] that the missing entries, with high probability, can be exactly restored by the convex program (2), as long as the target matrix $L_0$ is *low rank* and *incoherent* and the set $\Omega$ of locations corresponding to the observed entries is a set sampled *uniformly at random*. This pioneering work provides people several useful tools to investigate matrix completion and many other related problems. Those assumptions, including low-rankness, incoherence and uniform sampling, are now standard and widely used in the literatures, e.g., [14, 17, 22, 24, 28, 33, 34, 36]. In particular, the analyses in [17, 33, 36] show that, in terms of theoretical completeness, many nonconvex optimization based methods are as powerful as the convex program (2). Unfortunately, these theories still depend on the assumption of uniform sampling, and thus they cannot explain why there are many nonconvex methods which often do better than the convex program (2) in practice.

The missing data in practice, however, often occurs in a nonuniform and deterministic fashion instead of randomly. This is because the reason for an observation being missing usually depends on the unseen observations themselves. For example, in structure from motion and magnetic resonance imaging, typically the locations of the observed entries are concentrated around the main diagonal of a matrix[4], as shown in Figure 1. Moreover, as pointed out by [19, 21, 23], the incoherence condition is indeed not so consistent with the mixture structure of multiple subspaces, which is also a ubiquitous phenomenon in practice. There has been sparse research in the direction of nonuniform sampling, e.g., [18, 25–27, 31]. In particular, Negahban and Wainwright [26] studied the case of weighted entrywise sampling, which is more general than the setup of uniform sampling but still a special form of random sampling. Király et al. [18] considered deterministic sampling and is most related to this work. However, they had only established conditions to decide whether a particular entry of the matrix can be restored. In other words, the setup of [18] may not handle well the dependence among the missing entries. In summary, matrix completion still starves for practical theories and methods, although has attained considerable improvements in these years.

To break through the limits of the setup of random sampling, in this paper we introduce a new hypothesis called *isomeric condition*, which is a mixed concept that combines together the rank and coherence of $L_0$ with the locations and amount of the observed entries. In general, *isomerism* (noun

of isomeric) is a very mild hypothesis and only a little bit more strict than the well-known *oracle assumption*; that is, the number of observed entries in each row and column of $L_0$ is not smaller than the rank of $L_0$. It is arguable that the isomeric condition can hold even when the missing entries have irregular locations. In particular, it is provable that the widely used assumption of uniform sampling is *sufficient* to ensure isomerism, not necessary. Equipped with this new tool, isomerism, we prove a set of theorems pertaining to *missing data recovery* [35] and matrix completion. For example, we prove that, under the condition of isomerism, the exact solutions that identify the target matrix are included as critical points by the commonly used bilinear programs. This result helps to explain the widely observed phenomenon that there are many nonconvex methods performing better than the convex program (2) on real-world matrix completion tasks. In summary, the contributions of this paper mainly include:

$\diamond$ We invent a new hypothesis called isomeric condition, which provably holds given the standard assumptions of uniform sampling, low-rankness and incoherence. In addition, we also exemplify that the isomeric condition can hold even if the target matrix $L_0$ is not incoherent and the missing entries are placed irregularly. Comparing to the existing studies about nonuniform sampling, our setup is more general.

$\diamond$ Equipped with the isomeric condition, we prove that the exact solutions that identify $L_0$ are included as critical points by the commonly used bilinear programs. Comparing to the existing theories for nonconvex matrix completion, our theory is built upon a much weaker assumption and can therefore partially reveal the superiorities of nonconvex programs over the convex methods based on (2).

$\diamond$ We prove that the isomeric condition is *sufficient* and *necessary* for the column and row projectors of $L_0$ to be invertible given the sampling pattern $\Omega$. This result implies that the isomeric condition is *necessary* for ensuring that the minimal rank solution to (1) can identify the target $L_0$.

The rest of this paper is organized as follows. Section 2 summarizes the mathematical notations used in the paper. Section 3 introduces the proposed isomeric condition, along with some theorems for matrix completion. Section 4 shows some empirical results and Section 5 concludes this paper. The detailed proofs to all the proposed theorems are presented in the Supplementary Materials.

## 2 Notations

Capital and lowercase letters are used to represent matrices and vectors, respectively, except that the lowercase letters, $i, j, k, m, n, l, p, q, r, s$ and $t$, are used to denote some integers, e.g., the location of an observation, the rank of a matrix, etc. For a matrix $M$, $[M]_{ij}$ is its $(i, j)$th entry, $[M]_{i,:}$ is its $i$th row and $[M]_{:,j}$ is its $j$th column. Let $\omega_1$ and $\omega_2$ be two 1D index sets; namely, $\omega_1 = \{i_1, i_2, \cdots, i_k\}$ and $\omega_2 = \{j_1, j_2, \cdots, j_s\}$. Then $[M]_{\omega_1,:}$ denotes the submatrix of $M$ obtained by selecting the rows with indices $i_1, i_2, \cdots, i_k$, $[M]_{:,\omega_2}$ is the submatrix constructed by choosing the columns $j_1, j_2, \cdots, j_s$, and similarly for $[M]_{\omega_1,\omega_2}$. For a 2D index set $\Omega \subseteq \{1, 2, \cdots, m\} \times \{1, 2, \cdots, n\}$, we imagine it as a sparse matrix and, accordingly, define its "rows", "columns" and "transpose" as follows: The $i$th row $\Omega_i = \{j_1|(i_1, j_1) \in \Omega, i_1 = i\}$, the $j$th column $\Omega^j = \{i_1|(i_1, j_1) \in \Omega, j_1 = j\}$ and the transpose $\Omega^T = \{(j_1, i_1)|(i_1, j_1) \in \Omega\}$.

The special symbol $(\cdot)^+$ is reserved to denote the Moore-Penrose pseudo-inverse of a matrix. More precisely, for a matrix $M$ with Singular Value Decomposition (SVD)[5] $M = U_M \Sigma_M V_M^T$, its pseudo-inverse is given by $M^+ = V_M \Sigma_M^{-1} U_M^T$. For convenience, we adopt the conventions of using $\text{span}\{M\}$ to denote the linear space spanned by the columns of a matrix $M$, using $y \in \text{span}\{M\}$ to denote that a vector $y$ belongs to the space $\text{span}\{M\}$, and using $Y \in \text{span}\{M\}$ to denote that all the column vectors of a matrix $Y$ belong to $\text{span}\{M\}$.

Capital letters $U$, $V$, $\Omega$ and their variants (complements, subscripts, etc.) are reserved for left singular vectors, right singular vectors and index set, respectively. For convenience, we shall abuse the notation $U$ (resp. $V$) to denote the linear space spanned by the columns of $U$ (resp. $V$), i.e., the column space (resp. row space). The orthogonal projection onto the column space $U$, is denoted by $\mathcal{P}_U$ and given by $\mathcal{P}_U(M) = UU^T M$, and similarly for the row space $\mathcal{P}_V(M) = MVV^T$. The same

notation is also used to represent a subspace of matrices (i.e., the image of an operator), e.g., we say that $M \in \mathcal{P}_U$ for any matrix $M$ which satisfies $\mathcal{P}_U(M) = M$. We shall also abuse the notation $\Omega$ to denote the linear space of matrices supported on $\Omega$. Then the symbol $\mathcal{P}_\Omega$ denotes the orthogonal projection onto $\Omega$, namely,

$$[\mathcal{P}_\Omega(M)]_{ij} = \left\{ \begin{array}{ll} [M]_{ij}, & \text{if } (i,j) \in \Omega, \\ 0, & \text{otherwise.} \end{array} \right.$$

Similarly, the symbol $\mathcal{P}_\Omega^\perp$ denotes the orthogonal projection onto the complement space of $\Omega$. That is, $\mathcal{P}_\Omega + \mathcal{P}_\Omega^\perp = \mathcal{I}$, where $\mathcal{I}$ is the identity operator.

Three types of matrix norms are used in this paper, and they are all functions of the singular values: 1) The operator norm or 2-norm (i.e., largest singular value) denoted by $\|M\|$, 2) the Frobenius norm (i.e., square root of the sum of squared singular values) denoted by $\|M\|_F$ and 3) the nuclear norm or trace norm (i.e., sum of singular values) denoted by $\|M\|_*$. The only used vector norm is the $\ell_2$ norm, which is denoted by $\|\cdot\|_2$. The symbol $|\cdot|$ is reserved for the cardinality of an index set.

## 3 Isomeric Condition and Matrix Completion

This section introduces the proposed isomeric condition and a set of theorems for matrix completion. But most of the detailed proofs are deferred until the Supplementary Materials.

### 3.1 Isomeric Condition

In general cases, as aforementioned, matrix completion is an ill-posed problem. Thus, some assumptions are necessary for studying Problem 1.1. To eliminate the disadvantages of the setup of random sampling, we define and investigate a so-called *isomeric condition*.

#### 3.1.1 Definitions

For the ease of understanding, we shall begin with a concept called *k-isomerism* (or *k-isomeric* in adjective form), which could be regarded as an extension of low-rankness.

**Definition 3.1** (*k*-isomeric). *A matrix $M \in \mathbb{R}^{m \times l}$ is called k-isomeric if and only if any $k$ rows of $M$ can linearly represent all rows in $M$. That is,*

$$\text{rank}\left([M]_{\omega,:}\right) = \text{rank}\left(M\right), \forall \omega \subseteq \{1, 2, \cdots, m\}, |\omega| = k,$$

*where $|\cdot|$ is the cardinality of an index set.*

In general, $k$-isomerism is somewhat similar to *Spark* [37] which defines the smallest linearly dependent subset of the rows of a matrix. For a matrix $M$ to be $k$-isomeric, it is necessary that $\text{rank}(M) \leq k$, not sufficient. In fact, $k$-isomerism is also somehow related to the concept of *coherence* [4, 21]. When the coherence of a matrix $M \in \mathbb{R}^{m \times l}$ is not too high, the rows of $M$ will sufficiently spread, and thus $M$ could be $k$-isomeric with a small $k$, e.g., $k = \text{rank}(M)$. Whenever the coherence of $M$ is very high, one may need a large $k$ to satisfy the $k$-isomeric property. For example, consider an extreme case where $M$ is a rank-1 matrix with one row being 1 and everywhere else being 0. In this case, we need $k = m$ to ensure that $M$ is $k$-isomeric.

While Definition 3.1 involves all 1D index sets of cardinality $k$, we often need the isomeric property to be associated with a certain 2D index set $\Omega$. To this end, we define below a concept called $\Omega$-*isomerism* (or $\Omega$-*isomeric* in adjective form).

**Definition 3.2** ($\Omega$-isomeric). *Let $M \in \mathbb{R}^{m \times l}$ and $\Omega \subseteq \{1, 2, \cdots, m\} \times \{1, 2, \cdots, n\}$. Suppose that $\Omega^j \neq \emptyset$ (empty set), $\forall 1 \leq j \leq n$. Then the matrix $M$ is called $\Omega$-isomeric if and only if*

$$\text{rank}\left([M]_{\Omega^j,:}\right) = \text{rank}\left(M\right), \forall j = 1, 2, \cdots, n.$$

*Note here that only the number of rows in $M$ is required to coincide with the row indices included in $\Omega$, and thereby $l \neq n$ is allowable.*

Generally, $\Omega$-isomerism is less strict than $k$-isomerism. Provided that $|\Omega^j| \geq k, \forall 1 \leq j \leq n$, a matrix $M$ is $k$-isomeric ensures that $M$ is $\Omega$-isomeric as well, but not vice versa. For the extreme example where $M$ is nonzero at only one row, interestingly, $M$ can be $\Omega$-isomeric as long as the locations of the nonzero elements are included in $\Omega$.

With the notation of $\Omega^T = \{(j_1, i_1) | (i_1, j_1) \in \Omega\}$, the isomeric property could be also defined on the column vectors of a matrix, as shown in the following definition.

**Definition 3.3** ($\Omega/\Omega^T$-isomeric). *Let $M \in \mathbb{R}^{m \times n}$ and $\Omega \subseteq \{1, 2, \cdots, m\} \times \{1, 2, \cdots, n\}$. Suppose $\Omega_i \neq \emptyset$ and $\Omega^j \neq \emptyset$, $\forall i = 1, \cdots, m, j = 1, \cdots, n$. Then the matrix $M$ is called $\Omega/\Omega^T$-isomeric if and only if $M$ is $\Omega$-isomeric and $M^T$ is $\Omega^T$-isomeric as well.*

To solve Problem 1.1 without the imperfect assumption of missing at random, as will be shown later, we need to assume that $L_0$ is $\Omega/\Omega^T$-isomeric. This condition has excluded the unidentifiable cases where any rows or columns of $L_0$ are wholly missing. In fact, whenever $L_0$ is $\Omega/\Omega^T$-isomeric, the number of observed entries in each row and column of $L_0$ has to be greater than or equal to the rank of $L_0$; this is consistent with the results in [20]. Moreover, $\Omega/\Omega^T$-isomerism has actually well treated the cases where $L_0$ is of high coherence. For example, consider an extreme case where $L_0$ is 1 at only one element and 0 everywhere else. In this case, $L_0$ cannot be $\Omega/\Omega^T$-isomeric unless the nonzero element is observed. So, generally, it is possible to restore the missing entries of a highly coherent matrix, as long as the $\Omega/\Omega^T$-isomeric condition is obeyed.

### 3.1.2 Basic Properties

While its definitions are associated with a certain matrix, the isomeric condition is actually characterizing some properties of a space, as shown in the lemma below.

**Lemma 3.1.** *Let $L_0 \in \mathbb{R}^{m \times n}$ and $\Omega \subseteq \{1, 2, \cdots, m\} \times \{1, 2, \cdots, n\}$. Denote the SVD of $L_0$ as $U_0 \Sigma_0 V_0^T$. Then we have:*

1. *$L_0$ is $\Omega$-isomeric if and only if $U_0$ is $\Omega$-isomeric.*

2. *$L_0^T$ is $\Omega^T$-isomeric if and only if $V_0$ is $\Omega^T$-isomeric.*

*Proof.* It could be manipulated that

$$[L_0]_{\Omega^j,:} = ([U_0]_{\Omega^j,:})\Sigma_0 V_0^T, \forall j = 1, \cdots, n.$$

Since $\Sigma_0 V_0^T$ is row-wisely full rank, we have

$$\mathrm{rank}\left([L_0]_{\Omega^j,:}\right) = \mathrm{rank}\left([U_0]_{\Omega^j,:}\right), \forall j = 1, \cdots, n.$$

As a result, $L_0$ is $\Omega$-isomeric is equivalent to $U_0$ is $\Omega$-isomeric. In a similar way, the second claim is proved as well. $\square$

It is easy to see that the above lemma is still valid even when the condition of $\Omega$-isomerism is replaced by $k$-isomerism. Thus, hereafter, we may say that a space is isomeric ($k$-isomeric, $\Omega$-isomeric or $\Omega^T$-isomeric) as long as its basis matrix is isomeric. In addition, the isomeric property is subspace successive, as shown in the next lemma.

**Lemma 3.2.** *Let $\Omega \subseteq \{1, 2, \cdots, m\} \times \{1, 2, \cdots, n\}$ and $U_0 \in \mathbb{R}^{m \times r}$ be the basis matrix of a Euclidean subspace embedded in $\mathbb{R}^m$. Suppose that $U$ is a subspace of $U_0$, i.e., $U = U_0 U_0^T U$. If $U_0$ is $\Omega$-isomeric then $U$ is $\Omega$-isomeric as well.*

*Proof.* By $U = U_0 U_0^T U$ and $U_0$ is $\Omega$-isomeric,

$$\mathrm{rank}\left([U]_{\Omega^j,:}\right) = \mathrm{rank}\left(([U_0]_{\Omega^j,:})U_0^T U\right) = \mathrm{rank}\left(U_0^T U\right)$$
$$= \mathrm{rank}\left(U_0 U_0^T U\right) = \mathrm{rank}\left(U\right), \forall 1 \leq j \leq n.$$

$\square$

The above lemma states that, in one word, the subspace of an isomeric space is isomeric.

### 3.1.3 Important Properties

As aforementioned, the isometric condition is actually necessary for ensuring that the minimal rank solution to (1) can identify $L_0$. To see why, let's assume that $U_0 \cap \Omega^\perp \neq \{0\}$, where we denote by $U_0 \Sigma_0 V_0^T$ the SVD of $L_0$. Then one could construct a nonzero perturbation, denoted as $\Delta \in U_0 \cap \Omega^\perp$, and accordingly, obtain a feasible solution $\tilde{L}_0 = L_0 + \Delta$ to the problem in (1). Since $\Delta \in U_0$, we have $\mathrm{rank}(\tilde{L}_0) \leq \mathrm{rank}(L_0)$. Even more, it is entirely possible that $\mathrm{rank}(\tilde{L}_0) < \mathrm{rank}(L_0)$. Such a case is unidentifiable in nature, as the global optimum to problem (1) cannot identify $L_0$. Thus,

to ensure that the global minimum to (1) can identify $L_0$, it is essentially necessary to show that $U_0 \cap \Omega^\perp = \{0\}$ (resp. $V_0 \cap \Omega^\perp = \{0\}$), which is equivalent to the operator $\mathcal{P}_{U_0}\mathcal{P}_\Omega\mathcal{P}_{U_0}$ (resp. $\mathcal{P}_{V_0}\mathcal{P}_\Omega\mathcal{P}_{V_0}$) is invertible (see Lemma 6.8 of the Supplementary Materials). Interestingly, the isomeric condition is indeed a *sufficient* and *necessary* condition for the operators $\mathcal{P}_{U_0}\mathcal{P}_\Omega\mathcal{P}_{U_0}$ and $\mathcal{P}_{V_0}\mathcal{P}_\Omega\mathcal{P}_{V_0}$ to be invertible, as shown in the following theorem.

**Theorem 3.1.** *Let $L_0 \in \mathbb{R}^{m \times n}$ and $\Omega \subseteq \{1, 2, \cdots, m\} \times \{1, 2, \cdots, n\}$. Let the SVD of $L_0$ be $U_0\Sigma_0V_0^T$. Denote $\mathcal{P}_{U_0}(\cdot) = U_0U_0^T(\cdot)$ and $\mathcal{P}_{V_0}(\cdot) = (\cdot)V_0V_0^T$. Then we have the following:*

1. *The linear operator $\mathcal{P}_{U_0}\mathcal{P}_\Omega\mathcal{P}_{U_0}$ is invertible if and only if $U_0$ is $\Omega$-isomeric.*

2. *The linear operator $\mathcal{P}_{V_0}\mathcal{P}_\Omega\mathcal{P}_{V_0}$ is invertible if and only if $V_0$ is $\Omega^T$-isomeric.*

The necessity stated above implies that the isomeric condition is actually a very mild hypothesis. In general, there are numerous reasons for the target matrix $L_0$ to be isomeric. Particularly, the widely used assumptions of low-rankness, incoherence and uniform sampling are indeed *sufficient* (but not necessary) to ensure isomerism, as shown in the following theorem.

**Theorem 3.2.** *Let $L_0 \in \mathbb{R}^{m \times n}$ and $\Omega \subseteq \{1, 2, \cdots, m\} \times \{1, 2, \cdots, n\}$. Denote $n_1 = \max(m, n)$ and $n_2 = \min(m, n)$. Suppose that $L_0$ is incoherent and $\Omega$ is a 2D index set sampled uniformly at random, namely $\Pr((i, j) \in \Omega) = \rho_0$ and $\Pr((i, j) \notin \Omega) = 1 - \rho_0$. For any $\delta > 0$, if $\rho_0 > \delta$ is obeyed and $\mathrm{rank}(L_0) < \delta n_2/(c \log n_1)$ holds for some numerical constant $c$ then, with high probability at least $1 - n_1^{-10}$, $L_0$ is $\Omega/\Omega^T$-isomeric.*

It is worth noting that the isomeric condition can be obeyed in numerous circumstances other than the case of uniform sampling *plus* incoherence. For example,

$$\Omega = \{(1, 1), (1, 2), (1, 3), (2, 1), (3, 1)\} \text{ and } L_0 = \begin{bmatrix} 1 & 0 & 0 \\ 0 & 0 & 0 \\ 0 & 0 & 0 \end{bmatrix},$$

where $L_0$ is a $3 \times 3$ matrix with 1 at $(1, 1)$ and 0 everywhere else. In this example, $L_0$ is not incoherent and the sampling is not uniform either, but it could be verified that $L_0$ is $\Omega/\Omega^T$-isomeric.

## 3.2 Results

In this subsection, we shall show how the isomeric condition can take effect in the context of nonuniform sampling, establishing some theorems pertaining to *missing data recovery* [35] as well as matrix completion.

### 3.2.1 Missing Data Recovery

Before exploring the matrix completion problem, for the ease of understanding, we would like to consider a *missing data recovery* problem studied by Zhang [35], which could be described as follows: Let $y_0 \in \mathbb{R}^m$ be a data vector drawn form some low-dimensional subspace, denoted as $y_0 \in \mathcal{S}_0 \subset \mathbb{R}^m$. Suppose that $y_0$ contains some available observations in $y_b \in \mathbb{R}^k$ and some missing entries in $y_u \in \mathbb{R}^{m-k}$. Namely, after a permutation,

$$y_0 = \begin{bmatrix} y_b \\ y_u \end{bmatrix}, y_b \in \mathbb{R}^k, y_u \in \mathbb{R}^{m-k}. \tag{3}$$

Given the observations in $y_b$, we seek to restore the unseen entries in $y_u$. To do this, we consider the prevalent idea that represents a data vector as a linear combination of the bases in a given dictionary:

$$y_0 = Ax_0, \tag{4}$$

where $A \in \mathbb{R}^{m \times p}$ is a dictionary constructed in advance and $x_0 \in \mathbb{R}^p$ is the representation of $y_0$. Utilizing the same permutation used in (3), we can partition the rows of $A$ into two parts according to the indices of the observed and missing entries, respectively:

$$A = \begin{bmatrix} A_b \\ A_u \end{bmatrix}, A_b \in \mathbb{R}^{k \times p}, A_u \in \mathbb{R}^{(m-k) \times p}. \tag{5}$$

In this way, the equation in (4) gives that

$$y_b = A_bx_0 \quad \text{and} \quad y_u = A_ux_0.$$

As we now can see, the unseen data $y_u$ could be restored, as long as the representation $x_0$ is retrieved by only accessing the available observations in $y_b$. In general cases, there are infinitely many representations that satisfy $y_0 = Ax_0$, e.g., $x_0 = A^+ y_0$, where $(\cdot)^+$ is the pseudo-inverse of a matrix. Since $A^+ y_0$ is the representation of minimal $\ell_2$ norm, we revisit the traditional $\ell_2$ program:

$$\min_x \frac{1}{2} \|x\|_2^2, \quad \text{s.t.} \quad y_b = A_b x, \tag{6}$$

where $\| \cdot \|_2$ is the $\ell_2$ norm of a vector. Under some verifiable conditions, the above $\ell_2$ program is indeed *consistently successful* in a sense as in the following: For any $y_0 \in \mathcal{S}_0$ with an arbitrary partition $y_0 = [y_b; y_u]$ (i.e., arbitrarily missing), the desired representation $x_0 = A^+ y_0$ is the unique minimizer to the problem in (6). That is, the unseen data $y_u$ is exactly recovered by firstly computing the minimizer $x^*$ to problem (6) and then calculating $y_u = A_u x^*$.

**Theorem 3.3.** *Let $y_0 = [y_b; y_u] \in \mathbb{R}^m$ be an authentic sample drawn from some low-dimensional subspace $\mathcal{S}_0$ embedded in $\mathbb{R}^m$, $A \in \mathbb{R}^{m \times p}$ be a given dictionary and $k$ be the number of available observations in $y_b$. Then the convex program* (6) *is consistently successful, provided that $\mathcal{S}_0 \subseteq \text{span}\{A\}$ and the dictionary $A$ is $k$-isomeric.*

Unlike the theory in [35], the condition of which is unverifiable, our $k$-isomeric condition could be verified in finite time. Notice, that the problem of missing data recovery is closely related to matrix completion, which is actually to restore the missing entries in multiple data vectors simultaneously. Hence, Theorem 3.3 can be naturally generalized to the case of matrix completion, as will be shown in the next subsection.

### 3.2.2 Matrix Completion

The spirits of the $\ell_2$ program (6) can be easily transferred to the case of matrix completion. Following (6), one may consider Frobenius norm minimization for matrix completion:

$$\min_X \frac{1}{2} \|X\|_F^2, \quad \text{s.t.} \quad \mathcal{P}_\Omega(AX - L_0) = 0, \tag{7}$$

where $A \in \mathbb{R}^{m \times p}$ is a dictionary assumed to be given. As one can see, the problem in (7) is equivalent to (6) if $L_0$ is consisting of only one column vector. The same as (6), the convex program (7) can also exactly recover the desired representation matrix $A^+ L_0$, as shown in the theorem below. The difference is that we here require $\Omega$-isomerism instead of $k$-isomerism.

**Theorem 3.4.** *Let $L_0 \in \mathbb{R}^{m \times n}$ and $\Omega \subseteq \{1, 2, \cdots, m\} \times \{1, 2, \cdots, n\}$. Suppose that $A \in \mathbb{R}^{m \times p}$ is a given dictionary. Provided that $L_0 \in \text{span}\{A\}$ and $A$ is $\Omega$-isomeric, the desired representation $X_0 = A^+ L_0$ is the unique minimizer to the problem in* (7).

Theorem 3.4 tells us that, in general, even when the locations of the missing entries are interrelated and nonuniformly distributed, the target matrix $L_0$ can be restored as long as we have found a proper dictionary $A$. This motivates us to consider the commonly used bilinear program that seeks both $A$ and $X$ simultaneously:

$$\min_{A,X} \frac{1}{2} \|A\|_F^2 + \frac{1}{2} \|X\|_F^2, \quad \text{s.t.} \quad \mathcal{P}_\Omega(AX - L_0) = 0, \tag{8}$$

where $A \in \mathbb{R}^{m \times p}$ and $X \in \mathbb{R}^{p \times n}$. The problem above is bilinear and therefore nonconvex. So, it would be hard to obtain a strong performance guarantee as done in the convex programs, e.g., [4, 21]. Interestingly, under a very mild condition, the problem in (8) is proved to include the exact solutions that identify the target matrix $L_0$ as the critical points.

**Theorem 3.5.** *Let $L_0 \in \mathbb{R}^{m \times n}$ and $\Omega \subseteq \{1, 2, \cdots, m\} \times \{1, 2, \cdots, n\}$. Denote the rank and SVD of $L_0$ as $r_0$ and $U_0 \Sigma_0 V_0^T$, respectively. If $L_0$ is $\Omega / \Omega^T$-isomeric then the exact solution, denoted by $(A_0, X_0)$ and given by*

$$A_0 = U_0 \Sigma_0^{\frac{1}{2}} Q^T, X_0 = Q \Sigma_0^{\frac{1}{2}} V_0^T, \forall Q \in \mathbb{R}^{p \times r_0}, Q^T Q = \mathtt{I},$$

*is a critical point to the problem in* (8).

To exhibit the power of program (8), however, the parameter $p$, which indicates the number of columns in the dictionary matrix $A$, must be close to the true rank of the target matrix $L_0$. This is

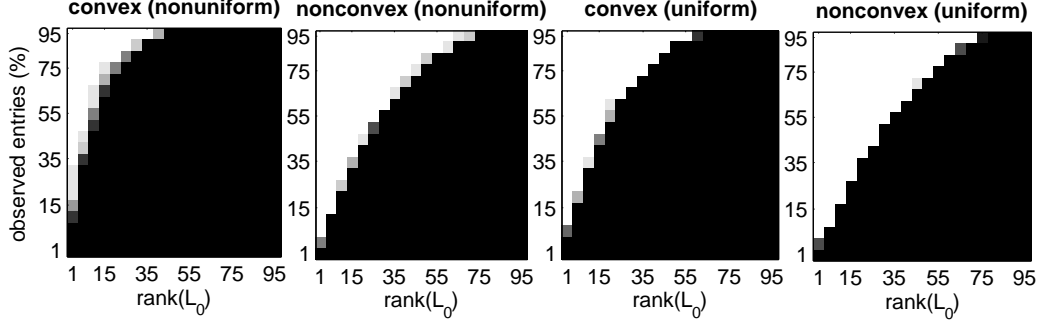

Figure 2: Comparing the bilinear program (9) ($p = m$) with the convex method (2). The numbers plotted on the above figures are the success rates within 20 random trials. The white and black points mean "succeed" and "fail", respectively. Here the success is in a sense that PSNR $\geq$ 40dB, where PSNR standing for peak signal-to-noise ratio.

impractical in the cases where the rank of $L_0$ is unknown. Notice, that the $\Omega$-isomeric condition imposed on $A$ requires

$$\text{rank}\,(A) \leq |\Omega^j|, \forall j = 1, 2, \cdots, n.$$

This, together with the condition of $L_0 \in \text{span}\{A\}$, essentially need us to solve a *low rank matrix recovery* problem [14]. Hence, we suggest to combine the formulation (7) with the popular idea of nuclear norm minimization, resulting in a bilinear program that jointly estimates both the dictionary matrix $A$ and the representation matrix $X$ by

$$\min_{A,X} \|A\|_* + \frac{1}{2}\|X\|_F^2, \text{ s.t. } \mathcal{P}_\Omega(AX - L_0) = 0, \tag{9}$$

which, by coincidence, has been mentioned in a paper about optimization [32]. Similar to (8), the program in (9) has the following theorem to guarantee its performance.

**Theorem 3.6.** *Let $L_0 \in \mathbb{R}^{m \times n}$ and $\Omega \subseteq \{1, 2, \cdots, m\} \times \{1, 2, \cdots, n\}$. Denote the rank and SVD of $L_0$ as $r_0$ and $U_0\Sigma_0 V_0^T$, respectively. If $L_0$ is $\Omega/\Omega^T$-isomeric then the exact solution, denoted by $(A_0, X_0)$ and given by*

$$A_0 = U_0 \Sigma_0^{\frac{2}{3}} Q^T, X_0 = Q\Sigma_0^{\frac{1}{3}} V_0^T, \forall Q \in \mathbb{R}^{p \times r_0}, Q^T Q = \mathtt{I},$$

*is a critical point to the problem in* (9).

Unlike (8), which possesses superior performance only if $p$ is close to $\text{rank}\,(L_0)$ and the initial solution is chosen carefully, the bilinear program in (9) can work well by simply choosing $p = m$ and using $A = \mathtt{I}$ as the initial solution. To see why, one essentially needs to figure out the conditions under which a specific optimization procedure can produce an optimal solution that meets an exact solution. This requires extensive justifications and we leave it as future work.

## 4 Simulations

To verify the superiorities of the nonconvex matrix completion methods over the convex program (2), we would like to experiment with randomly generated matrices. We generate a collection of $m \times n$ ($m = n = 100$) target matrices according to the model of $L_0 = BC$, where $B \in \mathbb{R}^{m \times r_0}$ and $C \in \mathbb{R}^{r_0 \times n}$ are $\mathcal{N}(0, 1)$ matrices. The rank of $L_0$, i.e., $r_0$, is configured as $r_0 = 1, 5, 10, \cdots, 90, 95$. Regarding the index set $\Omega$ consisting of the locations of the observed entries, we consider two settings: One is to create $\Omega$ by using a Bernoulli model to randomly sample a subset from $\{1, \cdots, m\} \times \{1, \cdots, n\}$ (referred to as "uniform"), the other is as in Figure 1 that makes the locations of the observed entries be concentrated around the main diagonal of a matrix (referred to as "nonuniform"). The observation fraction is set to be $|\Omega|/(mn) = 0.01, 0.05, \cdots, 0.9, 0.95$. For each pair of $(r_0, |\Omega|/(mn))$, we run 20 trials, resulting in 8000 simulations in total.

When $p = m$ and the identity matrix is used to initialize the dictionary $A$, we have empirically found that program (8) has the same performance as (2). This is not strange, because it has been proven in [16] that $\|L\|_* = \min_{A,X} \frac{1}{2}(\|A\|_F^2 + \|X\|_F^2)$, s.t. $L = AX$. Figure 2 compares the bilinear

program (9) to the convex method (2). It can be seen that (9) works distinctly better than (2). Namely, while handling the nonuniformly missing data, the number of matrices successfully restored by the bilinear program (9) is 102% more than the convex program (2). Even for dealing with the missing entries chosen uniformly at random, in terms of the number of successfully restored matrices, the bilinear program (9) can still outperform the convex method (2) by 44%. These results illustrate that, even in the cases where the rank of $L_0$ is unknown, the bilinear program (9) can do much better than the convex optimization based method (2).

## 5    Conclusion and Future Work

This work studied the problem of matrix completion with nonuniform sampling, a significant setting not extensively studied before. To figure out the conditions under which exact recovery is possible, we proposed a so-called isomeric condition, which provably holds when the standard assumptions of low-rankness, incoherence and uniform sampling arise. In addition, we also exemplified that the isomeric condition can be obeyed in the other cases beyond the setting of uniform sampling. Even more, our theory implies that the isomeric condition is indeed necessary for making sure that the minimal rank completion can identify the target matrix $L_0$. Equipped with the isomeric condition, finally, we mathematically proved that the widely used bilinear programs can include the exact solutions that recover the target matrix $L_0$ as the critical points; this guarantees the recovery performance of bilinear programs to some extend.

However, there still remain several problems for future work. In particular, it is unknown under which conditions a specific optimization procedure for (9) can produce an optimal solution that exactly restores the target matrix $L_0$. To do this, one needs to analyze the convergence property as well as the recovery performance. Moreover, it is unknown either whether the isomeric condition suffices for ensuring that the minimal rank completion can identify the target $L_0$. These require extensive justifications and we leave them as future work.

## Acknowledgment

We would like to thanks the anonymous reviewers and meta-reviewers for providing us many valuable comments to refine this paper.

## Footnotes

[4]This statement means that the observed entries are concentrated around the main diagonal after a permutation of the sampling pattern $\Omega$.

[5]In this paper, SVD always refers to skinny SVD. For a rank-$r$ matrix $M \in \mathbb{R}^{m \times n}$, its SVD is of the form $U_M \Sigma_M V_M^T$, where $U_M \in \mathbb{R}^{m \times r}, \Sigma_M \in \mathbb{R}^{r \times r}$ and $V_M \in \mathbb{R}^{n \times r}$.

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
