[Supplementary Material · nips_2017_appendix.pdf]

# 6 Supplemental Materials: Mathematical Proofs

This section shows the detailed proofs to the proposed theorems.

## 6.1 Basic Lemmas

The following lemma reveals the fact that the isomeric property is related to the invertibility of the sub-matrices of a basis matrix.

**Lemma 6.1.** *Let $\Omega \subseteq \{1, 2, \cdots, m\} \times \{1, 2, \cdots, n\}$ and $U_0 \in \mathbb{R}^{m \times r}$ be the basis matrix of a subspace embedded in $\mathbb{R}^m$. Denote the ith row of $U_0$ as $u_i^T$, i.e., $U_0 = [u_1^T; u_2^T; \cdots; u_m^T]$. Define $\delta_{ij}$ as*

$$\delta_{ij} = \begin{cases} 1, & \text{if } (i, j) \in \Omega, \\ 0, & \text{otherwise.} \end{cases} \tag{10}$$

*Then the matrices, $\sum_{i=1}^m \delta_{ij} u_i u_i^T, \forall 1 \leq j \leq n$, are all invertible if and only if $U_0$ is $\Omega$-isomeric.*

*Proof.* Note that

$$([U_0]_{\Omega^j,:})^T([U_0]_{\Omega^j,:}) = [\delta_{1j} u_1, \delta_{2j} u_2, \cdots, \delta_{mj} u_m] \begin{bmatrix} \delta_{1j} u_1^T \\ \delta_{2j} u_2^T \\ \vdots \\ \delta_{mj} u_m^T \end{bmatrix} = \sum_{i=1}^m (\delta_{ij})^2 u_i u_i^T = \sum_{i=1}^m \delta_{ij} u_i u_i^T.$$

Now, it is easy to see that $\sum_{i=1}^m \delta_{ij} u_i u_i^T$ is invertible is equivalent to that $([U_0]_{\Omega^j,:})^T([U_0]_{\Omega^j,:})$ is positive definite, which is further equivalent to that $\text{rank}\left([U_0]_{\Omega^j,:}\right) = \text{rank}(U_0), \forall j = 1, \cdots, n$. $\quad\square$

The next lemma will be used multiple times in the proof.

**Lemma 6.2.** *Let $\Omega \subseteq \{1, 2, \cdots, m\} \times \{1, 2, \cdots, n\}$ and $\mathcal{P}$ be an orthogonal projection onto some subspace of $\mathbb{R}^{m \times n}$. If $\|\mathcal{P}\mathcal{P}_\Omega^\perp \mathcal{P}\| < 1$ then $\mathcal{P}\mathcal{P}_\Omega \mathcal{P}$ is an invertible operator.*

*Proof.* Provided that $\|\mathcal{P}\mathcal{P}_\Omega^\perp \mathcal{P}\| < 1$, $\mathcal{I} + \sum_{i=1}^\infty (\mathcal{P}\mathcal{P}_\Omega^\perp \mathcal{P})^i$ is well defined. Also, notice that

$$\mathcal{P}\mathcal{P}_\Omega \mathcal{P} = \mathcal{P}(\mathcal{I} - \mathcal{P}_\Omega^\perp)\mathcal{P} = \mathcal{P}(\mathcal{I} - \mathcal{P}\mathcal{P}_\Omega^\perp \mathcal{P}).$$

Thus, for any $M \in \mathcal{P}$, the following holds:

$$\mathcal{P}\mathcal{P}_\Omega \mathcal{P}(\mathcal{I} + \sum_{i=1}^\infty (\mathcal{P}\mathcal{P}_\Omega^\perp \mathcal{P})^i)(M)$$

$$= \mathcal{P}(\mathcal{I} - \mathcal{P}\mathcal{P}_\Omega^\perp \mathcal{P})(\mathcal{I} + \sum_{i=1}^\infty (\mathcal{P}\mathcal{P}_\Omega^\perp \mathcal{P})^i)(M)$$

$$= \mathcal{P}(\mathcal{I} + \sum_{i=1}^\infty (\mathcal{P}\mathcal{P}_\Omega^\perp \mathcal{P})^i - \mathcal{P}\mathcal{P}_\Omega^\perp \mathcal{P} - \sum_{i=2}^\infty (\mathcal{P}\mathcal{P}_\Omega^\perp \mathcal{P})^i)(M)$$

$$= \mathcal{P}(M) = M.$$

In a similar way, it could be also verified that $(\mathcal{I} + \sum_{i=1}^\infty (\mathcal{P}\mathcal{P}_\Omega^\perp \mathcal{P})^i)\mathcal{P}\mathcal{P}_\Omega \mathcal{P}(M) = M$. As a consequence, $\mathcal{I} + \sum_{i=1}^\infty (\mathcal{P}\mathcal{P}_\Omega^\perp \mathcal{P})^i$ is the inverse operator of $\mathcal{P}\mathcal{P}_\Omega \mathcal{P}$. $\quad\square$

**Lemma 6.3.** *Let $\Omega \subseteq \{1, 2, \cdots, m\} \times \{1, 2, \cdots, n\}$ and $\mathcal{P}$ be an orthogonal projection onto some subspace of $\mathbb{R}^{m \times n}$. If $\|\mathcal{P}\mathcal{P}_\Omega^\perp \mathcal{P}\| < 1$ then $\mathcal{P} \cap \mathcal{P}_\Omega^\perp = \{0\}$.*

*Proof.* Suppose that $M \in \mathcal{P} \cap \mathcal{P}_\Omega^\perp$, i.e., $M = \mathcal{P}(M) = \mathcal{P}_\Omega^\perp(M)$. Then we have $M = \mathcal{P}\mathcal{P}_\Omega^\perp \mathcal{P}(M)$ and thus

$$\|M\|_F = \|\mathcal{P}\mathcal{P}_\Omega^\perp \mathcal{P}(M)\|_F \leq \|\mathcal{P}\mathcal{P}_\Omega^\perp \mathcal{P}\|\|M\|_F \leq \|M\|_F.$$

Since $\|\mathcal{P}\mathcal{P}_\Omega^\perp \mathcal{P}\| < 1$, the last equality above can hold only when $M = 0$. $\quad\square$

The following lemma is well-known.

**Lemma 6.4** (Lemma 11 of [29]). *For any matrices $M, N, W$ and $Z$ of consistent sizes, we have that*

$$\left\| \begin{bmatrix} M & N \\ W & Z \end{bmatrix} \right\|_* \geq \|M\|_*,$$

*where the equality can hold if and only if $N = 0$, $W = 0$ and $Z = 0$.*

*Proof.* By Lemma 11 of [29],

$$\left\| \begin{bmatrix} M & N \\ W & Z \end{bmatrix} \right\|_* \geq \|[M, N]\|_* \geq \|M\|_*.$$

The validity of the first equality requires that $W = 0$ and $Z = 0$. The second equality demands $N = 0$. □

## 6.2 Critical Lemmas

The following lemma (i.e., Theorem 3.1) has a critical role in the proof.

**Lemma 6.5** (Theorem 3.1). *Let $L_0 \in \mathbb{R}^{m \times n}$ and $\Omega \subseteq \{1, 2, \cdots, m\} \times \{1, 2, \cdots, n\}$. Let the SVD of $L_0$ be $U_0 \Sigma_0 V_0^T$. Denote $\mathcal{P}_{U_0}(\cdot) = U_0 U_0^T(\cdot)$ and $\mathcal{P}_{V_0}(\cdot) = (\cdot) V_0 V_0^T$. Then we have the following:*

1. *The linear operator $\mathcal{P}_{U_0} \mathcal{P}_\Omega \mathcal{P}_{U_0}$ is invertible if and only if $U_0$ is $\Omega$-isomeric.*

2. *The linear operator $\mathcal{P}_{V_0} \mathcal{P}_\Omega \mathcal{P}_{V_0}$ is invertible if and only if $V_0$ is $\Omega^T$-isomeric.*

*Proof.* The above two claims are proved in the same way, and thereby we only present the proof to first one. Since the operator $\mathcal{P}_{U_0} \mathcal{P}_\Omega \mathcal{P}_{U_0}$ is linear and $\mathcal{P}_{U_0}$ is a linear space of finite dimension, the sufficiency can be proved by showing that $\mathcal{P}_{U_0} \mathcal{P}_\Omega \mathcal{P}_{U_0}$ is an injection. That is, we need to prove that the following linear system has no nonzero solution:

$$\mathcal{P}_{U_0} \mathcal{P}_\Omega \mathcal{P}_{U_0}(M) = 0, \text{ s.t. } M \in \mathcal{P}_{U_0}.$$

Assume that $\mathcal{P}_{U_0} \mathcal{P}_\Omega \mathcal{P}_{U_0}(M) = 0$. Then we have

$$U_0^T \mathcal{P}_\Omega(U_0 U_0^T M) = 0.$$

Denote the $i$th row and $j$th column of $U_0$ and $U_0^T M$ as $u_i^T$ and $b_j$, respectively. That is, $U_0 = [u_1^T; u_2^T; \cdots; u_m^T]$ and $U_0^T M = [b_1, b_2, \cdots, b_n]$. Define $\delta_{ij}$ as in (10). Then the $j$th column of $U_0^T \mathcal{P}_\Omega(U_0 U_0^T M)$ is given by

$$U_0^T \begin{bmatrix} \delta_{1j} u_1^T b_j \\ \delta_{2j} u_2^T b_j \\ \vdots \\ \delta_{mj} u_m^T b_j \end{bmatrix} = (\sum_{i=1}^{m} \delta_{ij} u_i u_i^T) b_j.$$

By Lemma 6.1, the matrix $\sum_{i=1}^{m} \delta_{ij} u_i u_i^T$ is invertible. Hence, $U_0^T \mathcal{P}_\Omega(U_0 U_0^T M) = 0$ implies that

$$b_j = 0, \forall j = 1, \cdots, n,$$

i.e., $U_0^T M = 0$. By the assumption of $M \in \mathcal{P}_{U_0}$, $M = 0$.

It remains to prove the necessity. Assume that $U_0$ is not $\Omega$-isomeric. By Lemma 6.1, there exists $j$ such that the matrix $\sum_{i=1}^{m} \delta_{ij} u_i u_i^T$ is singular and therefore has a nonzero null space. So, there exists $M_1 \neq 0$ such that $U_0^T \mathcal{P}_\Omega(U_0 M_1) = 0$. Let $M = U_0 M_1$. Then we have $M \neq 0$, $M \in \mathcal{P}_{U_0}$ and

$$\mathcal{P}_{U_0} \mathcal{P}_\Omega \mathcal{P}_{U_0}(M) = 0.$$

This contradicts the assumption that $\mathcal{P}_{U_0} \mathcal{P}_\Omega \mathcal{P}_{U_0}$ is invertible. As a consequence, $U_0$ must be $\Omega$-isomeric. □

By Lemma 6.2, $\|\mathcal{P}_{U_0} \mathcal{P}_\Omega^\perp \mathcal{P}_{U_0}\| < 1$ also leads to the invertibility of $\mathcal{P}_{U_0} \mathcal{P}_\Omega \mathcal{P}_{U_0}$. So, according to Lemma 6.5, $\|\mathcal{P}_{U_0} \mathcal{P}_\Omega^\perp \mathcal{P}_{U_0}\| < 1$ should be related to the isomeric property. This is true, as shown in the following lemma.

**Lemma 6.6.** *Let $L_0 \in \mathbb{R}^{m \times n}$ and $\Omega \subseteq \{1, 2, \cdots, m\} \times \{1, 2, \cdots, n\}$. Let the SVD of $L_0$ be $U_0 \Sigma_0 V_0^T$. Denote $\mathcal{P}_{U_0}(\cdot) = U_0 U_0^T(\cdot)$ and $\mathcal{P}_{V_0}(\cdot) = (\cdot)V_0 V_0^T$. Then we have the following:*

*1. $\|\mathcal{P}_{U_0} \mathcal{P}_\Omega^\perp \mathcal{P}_{U_0}\| < 1$ if and only if $U_0$ is $\Omega$-isomeric.*

*2. $\|\mathcal{P}_{V_0} \mathcal{P}_\Omega^\perp \mathcal{P}_{V_0}\| < 1$ if and only if $V_0$ is $\Omega^T$-isomeric.*

*Proof.* The necessity could be proved by Lemma 6.2 and Lemma 6.5, and thereby we only need to prove the sufficiency. Denote $\delta_{ij}$ as in (10) and define a diagonal matrix $D_j$ as $D_j = \mathrm{diag}(\delta_{1j}, \delta_{2j}, \cdots, \delta_{mj}) \in \mathbb{R}^{m \times m}$. Then we have

$$([U_0]_{\Omega^j,:})^T([U_0]_{\Omega^j,:}) = U_0^T D_j^T D_j U_0 = U_0^T D_j U_0.$$

By Lemma 6.1, $U_0^T D_j U_0$ is positive definite and therefore has positive singular values. Also, we have $\|U_0^T D_j U_0\| \le \|D_j\| \le 1$. As a consequence,

$$\sigma_j \mathtt{I} \preccurlyeq U_0^T D_j U_0 \preccurlyeq \mathtt{I},$$

where $\sigma_j > 0$ is the minimal singular value of $U_0^T D_j U_0$. Denote the $j$th column of $\mathcal{P}_{U_0}(M)$ as $b_j$. Then we have

$$\begin{aligned} \|[\mathcal{P}_{U_0} \mathcal{P}_\Omega^\perp \mathcal{P}_{U_0}(M)]_{:,j}\|_2 &= \|U_0 U_0^T b_j - U_0(U_0^T D_j U_0)U_0^T b_j\|_2 \\ &= \|(\mathtt{I} - U_0^T D_j U_0)U_0^T b_j\|_2 \le \|(\mathtt{I} - U_0^T D_j U_0)\| \|U_0^T b_j\|_2 \\ &= (1 - \sigma_j)\|U_0^T b_j\|_2 = (1 - \sigma_j)\|b_j\|_2, \forall j = 1, \cdots, n, \end{aligned}$$

which implies that

$$\|\mathcal{P}_{U_0} \mathcal{P}_\Omega^\perp \mathcal{P}_{U_0}(M)\|_F^2 \le \sum_{j=1}^n (1 - \sigma_j)^2 \|b_j\|_2^2$$

$$\le (1 - \sigma_{min})^2 \|\mathcal{P}_{U_0}(M)\|_F^2,$$

where $\sigma_{min} = \min_j\{\sigma_j\} > 0$. Hence,

$$\|\mathcal{P}_{U_0} \mathcal{P}_\Omega^\perp \mathcal{P}_{U_0}\| \le 1 - \sigma_{min} < 1.$$

$\square$

Lemma 6.5 and Lemma 6.6 imply that $\|\mathcal{P}_{U_0} \mathcal{P}_\Omega^\perp \mathcal{P}_{U_0}\| < 1$ is a sufficient and necessary condition for $\mathcal{P}_{U_0} \mathcal{P}_\Omega \mathcal{P}_{U_0}$ to be invertible. In fact, this is true for any orthogonal projections, as stated in the following lemma.

**Lemma 6.7.** *Let $\Omega \subseteq \{1, 2, \cdots, m\} \times \{1, 2, \cdots, n\}$ and $\mathcal{P}$ be an orthogonal projection onto some $r$-dimensional subspace of $\mathbb{R}^{m \times n}$. Then the linear operator, $\mathcal{P}\mathcal{P}_\Omega \mathcal{P}$, is an invertible operator if and only if $\|\mathcal{P}\mathcal{P}_\Omega^\perp \mathcal{P}\| < 1$.*

*Proof.* The sufficiency has been proven by Lemma 6.2, and thus we only need to prove that $\|\mathcal{P}\mathcal{P}_\Omega^\perp \mathcal{P}\| < 1$ is necessary. Let $\mathrm{vec}(\cdot)$ denote the vectorization of a matrix formed by stacking the columns of the matrix into a single column vector. Suppose that the basis matrix associated with the operator $\mathcal{P}$ is given by $P \in \mathbb{R}^{mn \times r}, P^T P = \mathtt{I}$; namely,

$$\mathrm{vec}(\mathcal{P}(M)) = PP^T \mathrm{vec}(M), \forall M \in \mathbb{R}^{m \times n}.$$

Denote $\delta_{ij}$ as in (10) and define a diagonal matrix $D$ as

$$D = \mathrm{diag}(\delta_{11}, \delta_{21}, \cdots, \delta_{ij}, \cdots, \delta_{mn}) \in \mathbb{R}^{mn \times mn}.$$

Notice that

$$\mathcal{P}(M) = \mathcal{P}(\sum_{i=1}^m \sum_{j=1}^n \langle M, e_i e_j^T \rangle e_i e_j^T)$$

$$= \sum_{i=1}^m \sum_{j=1}^n \langle M, e_i e_j^T \rangle \mathcal{P}(e_i e_j^T),$$

where $e_i$ is the $i$th standard basis and $\langle \cdot \rangle$ denotes the inner product between two matrices. With this notation, it is easy to see that

$$[\text{vec}(\mathcal{P}(e_1 e_1^T)), \text{vec}(\mathcal{P}(e_2 e_1^T)), \cdots, \text{vec}(\mathcal{P}(e_m e_n^T))] = PP^T.$$

Similarly, we have

$$\mathcal{P}\mathcal{P}_\Omega \mathcal{P}(M) = \sum_{i=1}^m \sum_{j=1}^n \langle \mathcal{P}(M), e_i e_j^T \rangle (\delta_{ij} \mathcal{P}(e_i e_j^T)),$$

and thereby

$$\text{vec}(\mathcal{P}\mathcal{P}_\Omega \mathcal{P}(M)) = PP^T DPP^T \text{vec}(M).$$

For $\mathcal{P}\mathcal{P}_\Omega \mathcal{P}$ to be invertible, the matrix $P^T DP$ must be positive definite. To show this, let's assume that $P^T DP$ is singular. Then there exists a vector, $z \in \mathbb{R}^{mn}$, $z \neq 0$, that satisfies $P^T DPz = 0$. Let $\text{vec}(M) = Pz$. Then we have $PP^T DPP^T \text{vec}(M) = PP^T DPz = 0$. By $z \neq 0$, $\text{vec}(M) \neq 0$. Hence, there exists $M \in \mathcal{P}$ and $M \neq 0$ such that $\mathcal{P}\mathcal{P}_\Omega \mathcal{P}(M) = 0$. This contradicts the assumption that $\mathcal{P}\mathcal{P}_\Omega \mathcal{P}$ is invertible.

Denote the minimal singular value of $P^T DP$ as $\sigma_{min} > 0$. Then we have

$$\begin{aligned}
\|\mathcal{P}\mathcal{P}_\Omega^\perp \mathcal{P}(M)\|_F^2 &= \|\text{vec}(\mathcal{P}\mathcal{P}_\Omega^\perp \mathcal{P}(M))\|_2^2 \\
&= \|PP^T (\text{I} - D)PP^T \text{vec}(M)\|_2^2 \\
&= \|(\text{I} - P^T DP)P^T \text{vec}(M)\|_2^2 \\
&\leq (1 - \sigma_{min})^2 \|P^T \text{vec}(M)\|_2^2 \\
&= (1 - \sigma_{min})^2 \|\mathcal{P}(M)\|_F^2,
\end{aligned}$$

which gives that $\|\mathcal{P}\mathcal{P}_\Omega^\perp \mathcal{P}\| \leq 1 - \sigma_{min} < 1$. $\qquad\square$

The following lemma has been used in our discussions.

**Lemma 6.8.** *Let $\Omega \subseteq \{1, 2, \cdots, m\} \times \{1, 2, \cdots, n\}$ and $\mathcal{P}$ be an orthogonal projection onto some subspace of $\mathbb{R}^{m \times n}$. Then the operator, $\mathcal{P}\mathcal{P}_\Omega \mathcal{P}$, is invertible if and only if $\mathcal{P} \cap \mathcal{P}_\Omega^\perp = \{0\}$.*

*Proof.* The necessity has been proven by Lemma 6.7 and Lemma 6.3. So, it suffices to prove that $\mathcal{P} \cap \mathcal{P}_\Omega^\perp = \{0\}$ can lead to the invertibility of the operator $\mathcal{P}\mathcal{P}_\Omega \mathcal{P}$. Consider a nonzero matrix $M \in \mathcal{P}$. Then we have

$$\|M\|_F^2 = \|\mathcal{P}(M)\|_F^2 = \|\mathcal{P}_\Omega \mathcal{P}(M) + \mathcal{P}_\Omega^\perp \mathcal{P}(M)\|_F^2 = \|\mathcal{P}_\Omega \mathcal{P}(M)\|_F^2 + \|\mathcal{P}_\Omega^\perp \mathcal{P}(M)\|_F^2,$$

which gives that

$$\|\mathcal{P}\mathcal{P}_\Omega^\perp \mathcal{P}(M)\|_F^2 \leq \|\mathcal{P}_\Omega^\perp \mathcal{P}(M)\|_F^2 = \|M\|_F^2 - \|\mathcal{P}_\Omega \mathcal{P}(M)\|_F^2.$$

By $\mathcal{P} \cap \mathcal{P}_\Omega^\perp = \{0\}$, $\mathcal{P}_\Omega \mathcal{P}(M) \neq 0$. Thus,

$$\|\mathcal{P}\mathcal{P}_\Omega^\perp \mathcal{P}\|^2 \leq 1 - \inf_{\|M\|_F = 1} \|\mathcal{P}_\Omega \mathcal{P}(M)\|_F^2 < 1.$$

Again, by Lemma 6.7, the operator $\mathcal{P}\mathcal{P}_\Omega \mathcal{P}$ is invertible. $\qquad\square$

Consider a twinned problem of (7); namely,

$$\min_A \|A\|_*, \text{ s.t. } \mathcal{P}_\Omega(AX - L_0) = 0, \tag{11}$$

where $X \in \mathbb{R}^{p \times n}$ is supposed to be given. Similar to Theorem 3.4, we have the following lemma to guarantee the success of the above convex program.

**Lemma 6.9.** *Let $X \in \mathbb{R}^{p \times n}$ be a given matrix and $\Omega \subseteq \{1, 2, \cdots, m\} \times \{1, 2, \cdots, n\}$. If $L_0^T \in \text{span}\{X^T\}$ and $X^T$ is $\Omega^T$-isomeric then $A_0 = L_0 X^+$ is the unique minimizer to the problem in (11).*

*Proof.* Denote the SVDs of $L_0$, $X$ and $L_0X^+$ as $U_0\Sigma_0V_0^T$, $U_1\Sigma_1V_1^T$ and $U_2\Sigma_2V_2^T$, respectively. By $L_0^T \in \mathrm{span}\{X^T\}$, $V_0 = V_1V_1^TV_0$ and thus

$$A_0X = L_0X^+X = L_0V_1V_1^T = L_0.$$

That is, $A_0 = L_0X^+$ is feasible to (11). By standard convexity arguments [30], $A_0 = L_0X^+$ is an optimal solution to the problem in (11) if there exists a matrix $W$ (Lagrange multiplier) that obeys

$$\mathcal{P}_\Omega(W)X^T \in \partial\|L_0X^+\|_*,$$

where $\partial(\cdot)$ is the subgradient of a function. By Lemma 3.1, $V_1$ is $\Omega^T$-isomeric. Then Lemma 6.5 gives that $\mathcal{P}_{V_1}\mathcal{P}_\Omega\mathcal{P}_{V_1}$ is an invertible operator. Hence, we could define $W$ as

$$W = \mathcal{P}_{V_1}(\mathcal{P}_{V_1}\mathcal{P}_\Omega\mathcal{P}_{V_1})^{-1}(U_2V_2^T(X^T)^+).$$

With this notation, it can be calculated that

$$\begin{aligned}
\mathcal{P}_\Omega(W)X^T &= \mathcal{P}_{V_1}\mathcal{P}_\Omega(W)X^T \\
&= \mathcal{P}_{V_1}\mathcal{P}_\Omega\mathcal{P}_{V_1}(\mathcal{P}_{V_1}\mathcal{P}_\Omega\mathcal{P}_{V_1})^{-1}(U_2V_2^T(X^T)^+)X^T \\
&= U_2V_2^T(X^T)^+X^T = U_2V_2^TU_1U_1^T.
\end{aligned}$$

Since $(L_0X^+)^T \in \mathrm{span}\{X\}$, we have

$$V_2^TU_1U_1^T = V_2^T, \text{ i.e., } V_2 \subseteq U_1.$$

As a result,

$$\mathcal{P}_\Omega(W)X^T = U_2V_2^TU_1U_1^T = U_2V_2^T \in \partial\|L_0X^+\|_*,$$

which gives that $L_0X^+$ is an optimal solution to the convex optimization problem in (11).

It remains to prove that the optimal solution to (11) is unique. We shall consider a feasible perturbation $A = L_0X^+ + \Delta$ and show that the objective strictly increases whenever $\Delta \neq 0$. We have

$$0 = \mathcal{P}_\Omega(AX - L_0) = \mathcal{P}_\Omega(L_0X^+X - L_0 + \Delta X),$$

which gives that

$$\mathcal{P}_\Omega(\Delta X) = 0, \text{ i.e., } \Delta X \in \mathcal{P}_\Omega^\perp.$$

We also have $\Delta X \in \mathcal{P}_{V_1}$, and thus $\Delta X \in \mathcal{P}_{V_1} \cap \mathcal{P}_\Omega^\perp$. However, by Lemma 6.6 and Lemma 6.3, $\mathcal{P}_{V_1} \cap \mathcal{P}_\Omega^\perp = \{0\}$. As a consequence,

$$\Delta X = 0, \text{ i.e., } \Delta^T \in U_1^\perp \subseteq V_2^\perp,$$

where $U_1^\perp \subseteq V_2^\perp$ follows from $V_2 \subseteq U_1$. Then we have

$$\begin{aligned}
\|L_0X^+ + \Delta\|_* &= \|\begin{bmatrix} U_2^T \\ (U_2^\perp)^T \end{bmatrix}(L_0X^+ + \Delta)[V_2, V_2^\perp]\|_* \\
&= \left\|\begin{bmatrix} U_2^TL_0X^+V_2 & U_2^T\Delta V_2^\perp \\ 0 & (U_2^\perp)^T\Delta V_2^\perp \end{bmatrix}\right\|_*.
\end{aligned}$$

By Lemma 6.4,

$$\|L_0X^+ + \Delta\|_* \geq \left\|U_2^TL_0X^+V_2\right\|_* = \|L_0X^+\|_*,$$

where the equality can hold if and only if

$$U_2^T\Delta V_2^\perp = 0 \text{ and } (U_2^\perp)^T\Delta V_2^\perp = 0.$$

This gives that $\Delta V_2^\perp = 0$, i.e., $\Delta^T \in V_2$. However, we have already proven that $\Delta^T \in V_2^\perp$. Thus, $\|L_0X^+ + \Delta\|_*$ is strictly greater than $\|L_0X^+\|_*$ unless $\Delta = 0$. In other words, $A_0 = L_0X^+$ is the unique minimizer to (11). $\square$

## 6.3 Proof to Theorem 3.2

*Proof.* Let the SVD of $L_0$ be $U_0 \Sigma_0 V_0^T$. Denote $\mathcal{P}_{U_0}(\cdot) = U_0 U_0^T(\cdot)$, $\mathcal{P}_{V_0}(\cdot) = (\cdot) V_0 V_0^T$ and $\mathcal{P}_{T_0}(\cdot) = \mathcal{P}_{U_0}(\cdot) + \mathcal{P}_{V_0}(\cdot) - \mathcal{P}_{U_0} \mathcal{P}_{V_0}(\cdot)$. Suppose that $L_0$ is incoherent, $\mathrm{rank}\,(L_0) \le \delta n_2 / (c \log n_1)$ and $\Omega$ is a 2D index set sampled using a Bernoulli model,
$$\Pr((i,j) \in \Omega) = \rho_0 > \delta.$$
Under these conditions, Theorem 4.1 of [4] has proven that
$$\|\mathcal{P}_{T_0} \mathcal{P}_{\Omega}^{\perp} \mathcal{P}_{T_0}\| < 1 - \rho_0 + \delta < 1$$
holds with high probability. Note that
$$\begin{aligned}\mathcal{P}_{U_0} \mathcal{P}_{T_0}(M) &= \mathcal{P}_{U_0}(\mathcal{P}_{U_0}(M) + \mathcal{P}_{V_0}(M) - \mathcal{P}_{U_0}\mathcal{P}_{V_0}(M)) \\ &= \mathcal{P}_{U_0}(M)\end{aligned}$$
and
$$\begin{aligned}\mathcal{P}_{T_0} \mathcal{P}_{U_0}(M) &= \mathcal{P}_{U_0}\mathcal{P}_{U_0}(M) + \mathcal{P}_{V_0}\mathcal{P}_{U_0}(M) - \mathcal{P}_{U_0}\mathcal{P}_{V_0}\mathcal{P}_{U_0}(M) \\ &= \mathcal{P}_{U_0}(M).\end{aligned}$$
Hence,
$$\begin{aligned}\|\mathcal{P}_{U_0} \mathcal{P}_{\Omega}^{\perp} \mathcal{P}_{U_0}\| &= \|\mathcal{P}_{U_0} \mathcal{P}_{T_0} \mathcal{P}_{\Omega}^{\perp} \mathcal{P}_{T_0} \mathcal{P}_{U_0}\| \\ &\le \|\mathcal{P}_{T_0} \mathcal{P}_{\Omega}^{\perp} \mathcal{P}_{T_0}\| < 1.\end{aligned}$$
By Lemma 6.6, $U_0$ is $\Omega$-isometric. Then it follows from Lemma 3.1 that $L_0$ is $\Omega$-isometric. Similarly, it could be proved that $L_0^T$ is $\Omega^T$-isometric with high probability. $\qquad\square$

## 6.4 Proof to Theorem 3.3

*Proof.* By $y_0 \in \mathcal{S}_0 \subseteq \mathrm{span}\{A\}$, $y_0 = AA^+ y_0$. By $y_0 = [y_b; y_u]$ and $A = [A_b; A_u]$,
$$y_b = A_b A^+ y_0.$$
That is, $x_0 = A^+ y_0$ is a feasible solution to the problem in (6). Provided that $y_b \in \mathbb{R}^k$ and the dictionary matrix $A$ is $k$-isomeric, Definition 3.1 gives that
$$\mathrm{rank}\,(A_b) = \mathrm{rank}\,(A),$$
which implies that the rows of $A_b$ can linearly represent the rows of $A$, i.e.,
$$\mathrm{span}\{A_b^T\} = \mathrm{span}\{A^T\}.$$
Since $A^+ y_0 \in \mathrm{span}\{A^T\}$, it follows that there exists a dual vector $w \in \mathbb{R}^p$ obeying
$$A_b^T w = A^+ y_0, \text{ i.e., } A_b^T w \in \partial \frac{1}{2}\|A^+ y_0\|_2^2.$$
By standard convexity arguments [30], $x_0 = A^+ y_0$ is an optimal solution to (6). Since $\|\cdot\|_2^2$ is strongly convex, the optimal solution to (6) is unique. $\qquad\square$

## 6.5 Proof to Theorem 3.4

*Proof.* Denote the SVD of $A$ as $U\Sigma V$. By $L_0 \in \mathrm{span}\{A\}$, $AX_0 = AA^+ L_0 = UU^T L_0 = L_0$; that is, $X_0 = A^+ L_0$ is a feasible solution to (7). By Lemma 3.1 and Lemma 6.5, the operator $\mathcal{P}_U \mathcal{P}_\Omega \mathcal{P}_U$ is invertible. As a consequence, we could define a matrix $W$ as
$$W = \mathcal{P}_U (\mathcal{P}_U \mathcal{P}_\Omega \mathcal{P}_U)^{-1} ((A^T)^+ X_0).$$
Then it can be calculated that
$$\begin{aligned}A^T \mathcal{P}_\Omega(W) &= A^T \mathcal{P}_U \mathcal{P}_\Omega(W) \\ &= A^T \mathcal{P}_U \mathcal{P}_\Omega \mathcal{P}_U (\mathcal{P}_U \mathcal{P}_\Omega \mathcal{P}_U)^{-1} ((A^T)^+ X_0) \\ &= A^T (A^T)^+ X_0 = VV^T X_0 \\ &= X_0 \in \partial \frac{1}{2}\|X_0\|_F^2,\end{aligned}$$
where $VV^T X_0 = X_0$ is concluded from the fact that $X_0 = A^+ L_0 \in \mathrm{span}(A^T)$. Since $\|X\|_F^2$ is a strongly convex function of $X$, it follows form the standard convexity arguments [30] that $X_0 = A^+ L_0$ is the unique optimal solution to the problem in (7). $\qquad\square$

## 6.6 Proof to Theorem 3.5

*Proof.* Since $A_0 = U_0 \Sigma_0^{\frac{1}{2}} Q^T$ and $X_0 = Q \Sigma_0^{\frac{1}{2}} V_0^T$, we have the following: 1) $A_0 X_0 = L_0$; 2) $L_0 \in$ span$\{A_0\}$ and $A_0$ is $\Omega$-isomeric; 3) $L_0^T \in$ span$\{X_0^T\}$ and $X_0^T$ is $\Omega^T$-isomeric. By Theorem 3.4,

$$X_0 = Q \Sigma_0^{\frac{1}{2}} V_0^T = A_0^+ L_0 = \arg \min_X \|X\|_F^2, \text{ s.t. } \mathcal{P}_\Omega(A_0 X - L_0) = 0,$$

$$A_0 = U_0 \Sigma_0^{\frac{1}{2}} Q^T = L_0 X_0^+ = \arg \min_A \|A\|_F^2, \text{ s.t. } \mathcal{P}_\Omega(A X_0 - L_0) = 0.$$

Hence, $(A_0, X_0)$ is a critical point to the problem in (8). $\qquad\qquad\square$

## 6.7 Proof to Theorem 3.6

*Proof.* Since $A_0 = U_0 \Sigma_0^{\frac{2}{3}} Q^T$ and $X_0 = Q \Sigma_0^{\frac{1}{3}} V_0^T$, we have the following: 1) $A_0 X_0 = L_0$; 2) $L_0 \in$ span$\{A_0\}$ and $A_0$ is $\Omega$-isomeric; 3) $L_0^T \in$ span$\{X_0^T\}$ and $X_0^T$ is $\Omega^T$-isomeric. By Theorem 3.4,

$$X_0 = Q \Sigma_0^{\frac{1}{3}} V_0^T = A_0^+ L_0$$
$$= \arg \min_X \frac{1}{2} \|X\|_F^2, \text{ s.t. } \mathcal{P}_\Omega(A_0 X - L_0) = 0.$$

By Lemma 6.9,

$$A_0 = U_0 \Sigma_0^{\frac{2}{3}} Q^T = L_0 X_0^+$$
$$= \arg \min_A \|A\|_*, \text{ s.t. } \mathcal{P}_\Omega(A X_0 - L_0) = 0.$$

Hence, $(A_0, X_0)$ is a critical point to the problem in (9). $\qquad\qquad\square$