[Reviews · NeurIPS 2017]

Reviewer 1



The authors study matrix completion from a few entries. They propose a new proprty of (matrix,sampling pattern) which they term "Isomery". They show that the well known (incoherence,low rank,random sampling) assumption implies this condition. The authors then consider a nonconvex bilinear program for matrix completion. Under the Isomery assumption, they prove that the exact solution is a critical point of this program (with no assumptions on rank, it seems). Some empirical evidence are presented, whereby the nonconvex program is superior to the convex Nuclear Norm minimization method for a certain pattern of nonuniform sampling. The paper is relatively well written (see comments on English below). The results are novel as far as I can tell, and interesting. This work touches on fundamental aspects of matrix recovery, and can certainly lead to new research. There are a few major comments I'd be happy to see addressed. That said, I vote accept. Major comments: 1. The authors say nothing about the natural questions: Can the Isometry condition be verified in practice? Is it a reasonable assumption to make? 2. The authors discuss in the introduction reasons why random sampling may not be a reasonable assumption, with some real data examples. It would be nice if, for a real full data matrix, and a sampling pattern of the kind shown, the authors can verify the Isomry assumption. 3. The matrix recovery problem, as well as all the algorithms discussed in this paper, are invariant under row and col permutations. The authors discuss sampling patterns which are "concentrated around the main diagonal" - this condition is not invariant under permutations. Only invariant conditions make sense. 4. The main contributions claimed (l.79) are not backed up by the paper itself. Specifically, - it's unclear what is meant by "our theories are more flexible and powerful" - the second bullet point in the main contribution list (l.83) referes to "nonconvex programs" in general - The third bullet point (l.87) is not actuallty proved, and should not be advertised as a main contribution. The authors proved Theorem 3.1, but its connection to matrix completion is only vageuly described in the paragraph above Theorem 3.1. minor comments: - English throughout - Theorem 3.2: formally define "with high probability" - The word "theory" and "theories" in the title and throughout: I'm a non-native English speaker, yet I don't see why you use the word. - l.56: "incoherent assumption" -> "incoherence assumption" - l.75 "by the commonly used" -> "of the commonly used" - l.79 "hypothesis" -> condition - l.92, l.129 "proof processes" -> "proofs" - l.188 "where it denotes" -> "where we denote" - l.208 "that" -?? - Eq. (8) and Eq. (9) - maybe missing =0 on the right hand side? - Figure 2: label vertical axis - what do you mean by "proof of theory" (eg line 92)?